# Exploratory Analysis of Selected Components of the mTOR Pathway Reveals Potentially Crucial Associations with Childhood Malnutrition

**DOI:** 10.3390/nu14081612

**Published:** 2022-04-12

**Authors:** Parag Palit, Md Amran Gazi, Subhasish Das, Md Mehedi Hasan, Zannatun Noor, Jafrin Ferdous, Md Ashraful Alam, Sharika Nuzhat, Md Ridwan Islam, Mustafa Mahfuz, Rashidul Haque, Tahmeed Ahmed

**Affiliations:** 1Nutrition and Clinical Services Division, International Centre for Diarrhoeal Disease Research, Bangladesh (icddr,b), Dhaka 1212, Bangladesh; parag.palit@icddrb.org (P.P.); subhasish.das@icddrb.org (S.D.); md.hasan@icddrb.org (M.M.H.); jafrin.ferdous@icddrb.org (J.F.); mashraful@icddrb.org (M.A.A.); sharika.nuzhat@icddrb.org (S.N.); ridwan.islam@icddrb.org (M.R.I.); mustafa@icddrb.org (M.M.); tahmeed@icddrb.org (T.A.); 2Emerging Infections and Parasitology Laboratory, International Centre for Diarrhoeal Disease Research, Bangladesh (icddr,b), Dhaka 1212, Bangladesh; zannatun@icddrb.org (Z.N.); rhaque@icddrb.org (R.H.); 3Faculty of Medicine and Health Technology, University of Tampere, 33014 Tampere, Finland; 4Department of Global Health, University of Washington, Seattle, WA 98105, USA

**Keywords:** mTOR pathway, childhood malnutrition, gene expression

## Abstract

Dysregulations in the mammalian target of rapamycin (mTOR) pathway are associated with several human anomalies. We aimed to elucidate possible implications for potential aberrations in the mTOR pathway with childhood malnutrition. We analyzed the activity of phospho-mTORC1 and the expressions of several mTOR pathway genes, namely: *MTOR*, *TSC1*, *LAMTOR2*, *RPS6K1* and *RICTOR* from peripheral blood mononuclear cells isolated from venous blood of children suffering from different forms of malnutrition and compared them with those from healthy children. Significant reduction in the phosphorylation of mTORC1 was noted, as well as a decrease in expression of *LAMTOR2* gene and increase in *TSC1* gene expression were observed between malnourished children in comparison to the healthy children. The deregulation in the activity of the *TSC1* and *LAMTOR2* gene was significantly associated with all forms of childhood malnutrition. Our findings provide key insights into possible down-modulation in the overall activity of the mTOR pathway in childhood malnutrition. Further studies focusing on the analysis of a multitude of components involved in the mTOR pathway both at the gene and protein expression levels are required for conclusive evidence for the aforementioned proposition.

## 1. Introduction

Childhood malnutrition has been strongly associated with a number of acute illnesses and damages to a number of physiological activities, including restricted mental growth and development, compromised host immunity and contributes to about 45% of cases of worldwide childhood mortality [1]. Stunting is the most prevalent form of childhood malnutrition, characterized by a length for age Z score of less than −2 (LAZ < −2) [2] which affects an estimated 159 million children worldwide [3]. “At-risk of stunting” is considered if the length-for-age Z score falls between −1 and −2 standard deviation for the reference population [2,3,4]. On the other hand, it has been estimated that on a worldwide scale, more than 18 million children under the age of 5 are affected by the most extreme form of malnutrition, severe acute malnutrition (SAM), which is characterized either by a weight for length Z score of less than −3 (WLZ < −3) or by the presence of bilateral pedal edema, independent of anthropometric indices [2].

Recent evidence indicates that human growth is controlled by the master growth regulation pathway, known as the mammalian target of the rapamycin (mTOR) pathway [5]. The mTOR pathway represents a vital intracellular signaling cascade that monitors the availability of nutrients, mitogenic signals and cellular energy levels and is involved in the regulation of cell growth and proliferation [5,6]. The mTOR pathway operates at two distinct multi-protein complexes, namely: mTOR Complex1 (mTORC1) and mTOR Complex2 (mTORC2) [7] and has been reported to respond to the availability of circulating levels of certain essential amino acids, such as leucine and tryptophan [8]. Perturbations in the activity of the mTOR signaling pathway have been linked to numerous human malignancies, cardiovascular diseases, obesity, neurodegenerative diseases and metabolic disorders [7]. However, a number of abnormalities exist where the roles of the possible aberrations in the mTOR signaling pathway have not been clearly understood.

No comprehensive study has been conducted until now, which has envisaged establishing a possible association between the mTOR pathway with the various forms of childhood malnutrition. In this study, we aimed to explore a potential aberration in the mTOR pathway with the phenotype of childhood malnutrition. We attempted to compare the activity of several mTOR pathway components among malnourished children (stunted, at risk of stunting and SAM) in comparison to healthy children. These selected mTOR pathway components included: phospho-mTORC1/the activated form of mTORC1 [9]; *TSC complex subunit 1 (TSC1)*, which is involved in the negative regulation of the pathway [10]; *MTOR*, which encodes the crucial subunit component of mTORC1 [11]; *LAMTOR2*, which is reported to be involved in the positive regulation of the mTORC1 pathway [12]; *RPS6K1* gene, which encodes a downstream modulator of the mTOR pathway [13] and *RICTOR*, a gene encoding a crucial structural sub-unit of mTORC2 [14]. Stable baseline expression of all these mTOR pathway genes was found in healthy human PBMC samples, as confirmed by “The Human Protein Atlas” database [15]. We hypothesized that the activity of the mTOR pathway, including the core components and the genes involved in the positive regulation, will be repressed, whereas the activity of the genes involved in the negative modulation of the mTOR pathway will be elevated among the malnourished children, in comparison to the healthy children.

## 2. Materials and Methods

### 2.1. Study Design, Setting and Population

This study was a cross-sectional study conducted in a cross-sectional manner in the slum areas and adjacent localities of Bauniabadh, Mirpur, Dhaka and in the Dhaka Hospital of icddr,b between October 2018 to February 2019. The study participants enrolled from the community were aged between 8 and 18 months of either sex recruited into two malnourished cohorts comprising of: stunted (LAZ < −2) and children at risk of stunting (LAZ < −1 to −2). Healthy children (LAZ, WAZ, WLZ > −1 and having no bilateral pedal edema) of identical age groups were recruited from the same demographic area in the community and served as the controls. Age and sex matched children with SAM (WLZ < −3 or presence of bilateral pedal edema independent of anthropometric measurements) were enrolled exclusively from the Dhaka Hospital of icddr,b, following acute phase management and prior to the start of nutritional rehabilitation.

### 2.2. Enrolment of Study Participants and Collection of Socio-Demographic Data

The study participants were selected for enrolment into either of the four study cohorts based upon the anthropometric indices (LAZ, WAZ and WLZ). Individuals with a history of cardiovascular illness, tuberculosis, cancer, impaired neurological and cognitive development, congenital anomalies and any known chronic illness were excluded upon primary screening. Moreover, participants screened with an ongoing episode of diarrhea, history of persistent diarrhea in the past month or history of acute diarrhea in the past 7 days, incidences of recent viral hepatitis, acute respiratory infections and asthma also served as exclusion criteria, since these physiological conditions have been reported to perturb the of the activity of the mTOR pathway [6,7,8,9,10,11,12,13,14,15,16]. All medical history was examined by a study physician, followed by a thorough medical examination of the participants prior to enrolment. Figure 1 shows the flow chart highlighting the methodology used to screen and enroll participants in the study.

Measurement of length (to the nearest 0.1 cm) and weight (to the nearest 0.1 kg) of the study participants was conducted by using a portable infantometer and weighing scales (Seca, Hamburg, Germany). Head circumference (to the nearest 0.1 cm) was measured using measuring tape, while mild-upper arm circumference (to the nearest cm) was measured using MUAC tape (Maternova, Providence, RI, USA). All anthropometric measurements were conducted by trained research assistants and standardized protocols for anthropometric measurements described elsewhere were followed all throughout the study [17]. Evaluation of anthropometric indices was carried out using the WHO AnthroPlus software (v1.0.4). Assessment of the presence of bilateral pedal edema, which served as an inclusion criterion for the enrolment of participants into the SAM group, was conducted exclusively by experienced physicians working at the Dhaka Hospital of icddr,b. The SAM children were admitted to the Dhaka Hospital of icddr,b with diarrhea and/or other co-morbidities and received appropriate therapy for acute phase management as practiced in the hospital. After the acute phase management, these SAM children were transferred to the Nutrition Rehabilitation Unit (NRU), where they received an appropriate therapeutic diet for weight gain. In our study, we enrolled the SAM children following their acute phase management and prior to the start of nutritional intervention for weight gain.

Following enrolment of the study participants, our research assistants interviewed the parents or legal guardians of the participants to collect data on socio-demographic factors and feeding practices in structured questionnaires.

### 2.3. Sample Size

Owing to the very limited number of published studies addressing the potential aberration in the activity of the mTOR signaling pathway in human anomalies, the sample size for this proposed study was calculated using the data from a study performed on children of the age group of 4.2–10.8 years of either sex suffering from asthma [16]. In this study, a standard deviation of 14.29 pg/mL in the concentrations of activated mTORC1 was observed among the asthma patients. Using this standard deviation at an absolute error of 5 pg/mL and a cross sectional sampling method, the sample size was calculated as follows:

Sample size = (Standard normal variate)^2^ × (Standard deviation)^2^/(Absolute Error)^2^ [18], where the value of the standard normal variate or Z_1-α_ is fixed at 1.96 at an absolute error of 5 pg/mL. Using this formula, the sample size was calculated to 31 samples per study arm, thus making up a total of 124 participants distributed equally across the four study groups.

### 2.4. Sample Collection and Processing

After enrolment of the participants in the study, 5 mL of venous blood was extracted following appropriate aseptic techniques in EDTA tubes (S-Monovette, Sarstedt, Nümbrecht, Germany) by trained clinical staff and then transported to the laboratory under maintenance of proper cold chain. For the participants enrolled from the community, blood was collected following overnight fasting in an attempt to avoid potential post-prandial effects [19]. Blood was collected from the SAM participants after acute phase management following standardized protocols used in the Dhaka Hospital of icddr,b for the management of SAM [20].

In the laboratory, peripheral blood mononuclear cells (PBMC) were isolated from the blood by ficoll-hypaque density gradient centrifugation [21]. Following PBMC isolation, cells were diluted 20 times in PBS and cell counting was conducted under 40× magnification and cell viability was checked by Trypan blue staining [22]. Only samples with cell viability of 90% were considered for immediate cryopreservation using appropriate freezing medium (90% FBS + 10% DMSO) in liquid nitrogen. For subsequent laboratory analyses, cryopreserved cells were thawed following procedures described elsewhere [23].

### 2.5. Extraction of Total Intracellular Proteins

Thawed PBMC samples were lysed using commercially available lysis buffer (RayBiotech, Peachtree Corners, GA, USA), reconstituted with protease and phosphatase inhibitors (Sigma-Aldrich, St. Louis, MO, USA). The volume of lysis buffer that was added to the thawed cells was adjusted according to the viable cell count obtained after thawing of PBMC samples, following which cells were incubated at 4 °C for 30 min with gentle shaking. After this incubation, centrifugation was performed at 10,000× *g* for 20 min and the supernatant (cell lysate) was collected and stored at −80 °C temperature until subsequent analysis.

### 2.6. Assessment of Activity of Phospho-mTORC1 Using ELISA

Cell lysates stored at −80 °C temperature was thawed on ice and were subject to 1:5 dilution using assay diluent supplied by the manufacturer prior to being run in duplicate wells of a pre-coated phospho-mTORC1 ELISA plate (RayBiotech, Peachtree Corners, GA, USA). Subsequent procedures for the ELISA were conducted as per the instructions and guidelines from the manufacturer. Optical density (OD) readings were taken at 450 nm and the mean OD value for each sample at 450 nm was calculated and used for the subsequent analyses, as described elsewhere [24].

### 2.7. Isolation of Total RNA and cDNA Synthesis by Reverse Transcriptase PCR (RT-PCR)

Extraction of RNA from thawed PBMC was performed using the RNeasy Mini kit (Qiagen, Germantown, MD, USA), according to the manufacturer’s instructions with minor adjustments. The concentration and purity of RNA were checked using Thermo Scientific NanoDrop ND-1000 (Waltham, MA, USA). The RNA samples with 260/280 wavelength ratios between 1.9 and 2.1 and 260/230 wavelength ratios more than 2.0 were considered as good quality samples [25]. The RNA samples were then aliquoted in RNase-free polypropylene tubes (Ambion™, Austin, TX, USA) and stored at −80 °C temperature, until further use. cDNA synthesis was performed to reverse transcribe all messenger RNAs (mRNAs) using Thermo Scientific™ RevertAid™ RT Kit according to the manufacturer’s instructions with oligodT primers; 3 μg of RNA was used for the reverse transcriptase reaction to produce the cDNA, scaled to a final reaction volume of 20 μL in Mastercycler (Eppendorf, Hamburg, Germany) using a pulse RT reaction cycle (protocol: 25 °C for 5 min; 42 °C for 30 min and 70 °C for 5 min). The cDNA was stored at −80 °C until subsequent analyses.

### 2.8. Expression Analysis of Selected mTOR Pathway Genes

Real-time PCR was performed in a 96-well CFX 96 Touch™ Real Time Detection System (BioRad, Irvine, CA, USA) with iTaq Universal SYBR Green Supermix (BioRad, Irvine, CA, USA) for the assessment of the expression of the selected mTOR pathway genes in the four cohorts (stunted, at risk of stunting SAM, healthy control), for gene expression analysis of each selected mTOR pathway genes, namely: *MTOR*, *TSC1*, *LAMTOR2*, *RPS6K1* and *RICTOR*. The primer sequences used for the gene expression studies have been shown in Appendix A. The total reaction volume of 10 μL consisted of 5 μL of SYBR Green Supermix, 0.175 μL of each primer (forward and reverse), 2 μL of the template (from 1:5 dilution of cDNA with DNase/RNase free water) and 2.65 μL of DNase/RNase free water. The following protocol was used: one cycle for 2 min at 95 °C for initial polymerase activation step; then 40 cycles of 95 °C for 5 s for template denaturation, 15 s at the optimum temperature for each gene-specific primer for annealing; followed by a 60 °C for 30 s for extension and fluorescence measurement and a subsequent melt curve. For internal control, Glyceraldehyde-3-phosphate dehydrogenase (*GAPDH*) was used as the reference gene and the amplification protocol followed was the same as that used for the expression analysis of mTOR pathway genes. Results were quantified using CFX connect software (BioRad, Irvine, CA, USA) and the 2^−ΔΔCt^ method [26] was used to calculate the fold change of expression of each of the selected mTOR pathway genes with respect to the reference gene (*GAPDH*) in each of the four study cohorts.

### 2.9. Statistical Analysis

All statistical analyses were conducted using STATA 13.0 (college station, Texas, TX, USA) and GraphPad Prism version 8.0 (GraphPad Software, Inc., San Diego, CA, USA). Owing to the overall skewed nature of the data, non-parametric tests were used to compare the variables between the study groups. Mann–Whitney U test was used to compare the optical density for p-mTORC1 and gene expression data of the aforementioned genes from each of the three malnourished groups (stunted, at risk of stunting and SAM) with that of the healthy control group. Spearman Rank Correlation was conducted to assess the correlation between the anthropometric indices of LAZ, WAZ and WLZ with the activity of phospho-mTORC1 and gene expression data of the selected mTOR pathway genes. A *p*-value of less than 0.05 was considered to be statistically significant.

Multivariate logistic regression was performed to quantify the association of the optical density for phospho-mTORC1 and gene expression data of the selected mTOR pathway genes with the three malnourished phenotypes of stunting, at risk of stunting and SAM while using the healthy control group as reference. Odds ratio with the 95% CI along with the *p*-value was expressed for each of the analyzed explanatory variables (optical density for phospho-mTORC1 and normalized relative gene expression data of the selected mTOR pathway genes) in order to compare between the outcome variables, constituting the three malnourished groups. All variables with a *p*-value less than 0.2 from the bivariate analysis were adjusted in the final multivariate logistic regression model and other variables that are already reported to be associated with childhood malnutrition [27,28].

## 3. Results

### 3.1. Socio-Demographic, Anthropometric and Feeding Data of the Study Participants

A total of 124 participants equally distributed across the four study groups were enrolled. The proportion of male participants was the highest in the “at risk of stunting” group and lowest in the healthy control group. The stunted children were comparatively more underweight (WAZ: −2.42 vs. −1.17) and wasted (WLZ: −1.57 vs. −0.45) compared to the children at risk of stunting. Among the SAM children enrolled from the Dhaka Hospital of icddr,b, only one participant had bilateral pedal edema. Table 1 illustrates the summary of the socio-demographic, anthropometric and nutritional data from the study participants of the four study cohorts.

The duration of exclusive breastfeeding among the children at risk of stunting was comparatively lower than that in the other three study groups, while the proportion of children currently breastfeeding in the SAM cohort was comparatively lower than that in the other groups. The proportion of children currently being formula fed, fed with rice powder/suji and given cow/goat’s milk was also higher in the SAM group, compared to that reported in the other three groups.

### 3.2. Assessment of Activity Phospho-mTORC1 and Correlation with Anthropometric Outcomes

The activity of phospho-mTORC1 as indicated by the OD values at 450 nm, was higher in the healthy control group compared to that in the malnourished groups. The optical density (OD) values for phospho-mTORC1 were higher for the participants in the healthy control group in comparison to that obtained from the PBMC samples of participants in each of the three malnourished arms (Appendix A). Figure 2 shows the comparison of the OD values for phospho-mTORC1 at 450 nm for each of the malnourished study arms with that of the healthy control group.

The OD values for p-mTORC1 at 450 nm for the SAM group were found to be significantly lowered in comparison to that from the healthy control group. No statistically significant difference in OD values for p-mTORC1 at 450 nm for the stunted and for the “at risk of stunting” group was observed in comparison to the healthy control group. Moreover, the activity of p-mTORC1 (as denoted by OD values at 450 nm) was found to have significant statistical correlations with all the anthropometric indices (LAZ, WAZ and WLZ) as shown in Figure 3. The strength of correlation of the activity of p-mTORC1 was the highest with WAZ and the lowest with LAZ.

Table 2 illustrates the statistical comparison of the normalized gene expression values for the aforementioned mTOR pathway genes between each of the three malnourished arms with the healthy control group. For the stunted group, the expression of *MTOR* and *LAMTOR2* was found to be significantly lower, while the expression of the *TSC1* gene was found to be significantly higher in comparison to that of the healthy control group. Consequently, the expression of the *RPS6K1* gene was found to be increased and that of the *RICTOR* gene was reduced for the stunted group in comparison to the healthy control group, though no statistical significance was found for the difference in the expression of these genes among the aforementioned groups.

Similar to the stunted group, the expression of the *MTOR* and *LAMTOR2* gene was found to be significantly reduced and that of TSC1 was found to be significantly elevated for the “at risk of stunting” group in comparison to the expression of these genes in the healthy control group. Additionally, no statistical significance was found in the expression of the *RPS6K1* gene between the children from “at risk of stunting” group compared to the children from the healthy control group. However, the expression of the *RICTOR* gene was found to be significantly reduced in “at risk of stunting” group in comparison to the healthy control group. In the SAM group, the expression of the *LAMTOR2* gene was found to be significantly lowered in comparison to that in the healthy control group. Subsequently, the expression of the *TSC1* gene was found to be significantly elevated in the SAM group in comparison to the healthy control group.

Consequently, correlation analyses revealed a significant positive correlation between the expression of the *LAMTOR2* gene with the anthropometric measurements of LAZ, WAZ and WLZ. A significant negative correlation was observed between the expression of the *TSC1* gene and the anthropometric measurement of LAZ. Figure 4 illustrates the aforementioned significant correlations between the gene expressions and the anthropometric indices. All the results from the correlation analysis between the selected mTOR pathway genes and different anthropometric measurements (LAZ, WAZ, and WLZ) have been summarized in Appendix A.

### 3.3. Association of the Assessed mTOR Pathway Components with Different Forms of Childhood Malnutrition

The association between the analyzed mTOR pathway components with the different forms of childhood malnutrition, as determined from multivariate logistic regression analysis has been shown in Table 3.

Stunting was found to have statistically significant negative associations with p-mTORC1 protein expression and *LAMTOR2* gene expression and a statistically significant positive association with *TSC1* gene expression. The expression of the *MTOR* gene was found to be negatively associated with childhood stunting but was not found to be statistically significant. Statistically significant negative associations were found between the expression of the *MTOR* gene and *LAMTOR2* gene with the nutritional status of “at risk stunting”. On the other hand, a statistically significant positive association was observed between the expression of the *TSC1* gene and the phenotype of at risk of stunting. Negative associations were found between p-mTORC1 expression and *RICTOR* gene expression; however, these associations were not statistically significant. Severe acute malnutrition (SAM) was found to have statistically significant negative associations with phospho-mTORC1 protein expression and with *TSC1* and *LAMTOR2* gene expression. Negative associations were observed between the expression of the *MTOR* gene and *RICTOR* gene with SAM; although not statistically significant.

## 4. Discussion

Despite, a plethora of attempts and extensive intervention strategies being undertaken, the prevalence of some of the most chronic forms of childhood malnutrition, such as stunting, has not reduced at the desired rate [29]. Several studies involving micronutrient interventions have failed to reduce childhood stunting [30,31,32]. Henceforth, in recent literature, malnutrition has not been simply associated with a lack of availability of food, but rather with a complex interplay of intra- and intergenerational factors [33]. Other reports have thus indicated a possible involvement of intracellular signaling pathways with the development of the phenotype of childhood malnutrition [34]. However, there is a distinct lack of literature that provides evidential proof regarding the potential role of the mTOR pathway in childhood malnutrition. Our study has shown the consistent differences in expression of several key components of the mTOR pathway, including phospho-mTORC1, *MTOR*, *TSC1* and *LAMTOR2* among malnourished children compared to that observed among healthy children. In addition, significant correlations were observed between the activity of phospho-mTORC1, *LAMTOR2* and *TSC1* with the anthropometric indices of LAZ, WAZ and WLZ. The expression of several of the mTOR pathway components analyzed in this study has been found to have a statistical association with the different forms of childhood malnutrition. To the best of our knowledge, this is the first study that has aimed to study the mTOR pathway, with extensive focus on the expressions of different key mTOR pathway components in the context of childhood malnutrition.

Chronic forms of malnutrition, such as stunting, are associated with environmental enteric dysfunction (EED), which is characterized by blunting of the intestinal villi and inflammation of the lamina propria [35]. The use of duodenal tissue biopsy samples would have been to study the molecular intricacies surrounding the phenotype of childhood malnutrition. However, the acquisition of such biological samples involves the use of the invasive medical procedure of upper GI endoscopy, which would not have been ethical in accordance with the design and methodology of our study. On the other hand, results from a number of studies involving the use of PBMC samples to study gene expression revealed that PBMCs provide an accurate reflection of the environment in the hepatic and intestinal tissues, with transcriptomics data supporting such observations [36]. Subsequent studies have used PBMC samples have been used to study gene expression profiles in celiac disease, which is accompanied by intestinal inflammation mirroring the condition as observed in EED [37].

Consequently, elevation in the levels of pro-inflammatory cytokines, such as IL-6, TNF-α and IL-1β has been observed among malnourished children [38,39], leading to decreased circulating levels of known inducers of the mTOR pathway, such as IGF-1 and growth hormone, through the modulation of the JAK-STAT signaling pathway [40]. On the other hand, dietary habits have also been reported to result in chronic forms of childhood malnutrition [41]. Poor dietary quality has been reported to be also linked with chronic forms of childhood malnutrition, whereby stunted children in sub-Saharan Africa have been reported to have lower circulating levels of certain essential amino acids [42]. Such findings can be implicated towards the consumption of a maize (*Zea mays*) based stable diet, which has been reported to be a poor source of the essential amino acid tryptophan [43]. Low circulating levels of tryptophan have also been reported among Bangladeshi stunted children [44], whereas other study findings have revealed positive associations between increased levels of essential amino acids in circulation and improved growth outcomes among Bangladeshi children [45]. On the other hand, previous studies have reported that the mTOR pathway is up-regulated by certain branched-chain amino acids, such as tryptophan, leucine, isoleucine and arginine as well as by nutrients, such as glucose, lipids and vitamin D [46,47].

Concurrent studies have reported the mTOR pathway to be exclusively sensitive to the availability of essential amino acids [48]. Deficiency in the levels of circulating essential amino acids has been shown to result in a down-modulation of mTORC1 activity, leading to reduced protein and lipid synthesis and activation of autophagy [49,50]. This availability of essential amino acids is sensed by another intracellular signaling pathway, known as the General control non-depressible 2 (GCN-2) pathway [51,52]. The GCN-2 pathway, in turn, acts as an antagonist to the mTOR pathway, leading to a reduction in protein synthesis, cell growth and proliferation [53]. Thus, the deregulation of certain key components of the mTOR pathway in childhood malnutrition may be attributed to a combinatory effect of increased levels of pro-inflammatory cytokines leading to reduced levels of IGF-1 and growth hormone and a deficiency of essential amino acids in circulation. Our study findings show that the activity of phospho-mTORC1, which is the activated form of mTORC1 [54], was decreased in the three malnourished groups (stunted, at risk of stunting and SAM) compared to that found among the healthy control children (Figure 2). This finding was strengthened by reduced expressions of the *MTOR* gene (Table 2). Subsequently, the activity of the *TSC1* gene, known to code for a well reported negative modulator of the mTORC1 pathway [10] was found to be increased among the malnourished children in comparison to the healthy children (Table 2). Moreover, the expression of *LAMTOR2*, reported to be a code for a vital positive modulator of the mTORC1 pathway, [12] was found to be decreased among the malnourished children in comparison to that observed among the healthy children (Table 2).

Recent works have indicated potential deregulation in the mTORC1 pathway with childhood stunting, possibly through disruptions in the chondral plate development and subsequent perturbations in bone growth [55]. Our study involving the down-modulation of the various components of the mTOR pathway among stunted children, including the significant positive correlation between p-mTORC1 with LAZ and negative correlations in the activity of the *TSC1* gene with LAZ, thus corroborate with such previous reports of disruptions in bone growth due to deregulations in the mTOR pathway, ultimately leading to linear growth faltering among the children. On the other hand, the mTOR pathway has been reported to control the processes of protein synthesis and lipogenesis [5,6]. In our study we report a positive correlation between p-mTORC1 and the activity of *LAMTOR2* with the anthropometric index of WAZ, indicating that a downregulation of these components of the mTOR pathway could result in reduced protein and lipid synthesis, resulting in reduced muscle mass and reduced ponderal growth.

The activity of *RPS6K1*, a gene known to encode a downstream modulator of mTORC1 [13] was found to be up-regulated among the malnourished children. This may potentially indicate the roles of other key modulators of the mTORC1 pathway, such as Sestrins [56]. Such interactions may have resulted in the aforementioned deregulation in the expression of the *RPS6K1* gene as determined from our analysis (Table 2). On the other hand, consistently reduced expressions of the *RICTOR* gene among the malnourished group with respect to that among the healthy control group indicate a potential role of the mTORC2 pathway with childhood malnutrition. The mTORC2 pathway has been implicated to exhibit crucial roles in the maintenance of cytoskeletal organization and in the mediation of autophagy, in a manner that is independent of the mTORC1 pathway [57]. Although crosstalk between the mTORC1 and mTORC2 has been reported, the mediators of such processes have not been elucidated to date [57]. Henceforth, we hypothesize that cues, such as pro-inflammatory cytokines, growth hormones and essential amino acids, which have been reported to be deregulated in several forms of childhood malnutrition may also act as upstream modulators of mTORC2, resulting in a down expression of the mTORC2 pathway.

### Limitations of the Study

Despite the findings of the study, that associate certain key components of the mTOR pathway with childhood malnutrition, there are a number of limitations to our findings. The use of biopsy samples from the upper GI tract or the epiphyseal plate could have been a more suitable representative to study the potential deregulations in the mTOR pathway in childhood malnutrition. However, in our study, we could not obtain such biopsy samples since that would have required the use of invasive protocols. We could not obtain blood from the SAM participants following overnight fasting, due to strict hospital guidelines regarding the management of diet for the SAM children; thereby, the issue of post-prandial effects may have been involved for the SAM children. Consequently, the mTOR pathway is an intricate intracellular signaling cascade involving the activity of a plethora of genes, transcription factors and upstream and downstream modulators. In our study, we had only investigated the expression of a limited number of mTOR pathway components, whereas components of other intracellular pathways known to be connected with the mTOR pathway were not studied. Lack of data regarding the protein expression hinder us from having conclusive evidence regarding possible deregulations of the mTOR pathway in childhood malnutrition, since post-transcriptional and post-translational modifications of the products of the gene expression may occur. Our study did not involve investigations into the known upstream modulators of the mTOR pathway, including IGF-1, growth hormone, pro-inflammatory cytokines, biomarkers of systemic inflammation and oxidative stress and levels of essential amino acids and branched chain amino acids from the biological samples from our study participants. Additionally, the findings of this study have been obtained from the pediatric Bangladeshi population and may not necessarily be extrapolated for other pediatric populations, owing to differences in epigenetic influences on gene expression.

## 5. Conclusions

Despite the aforementioned limitations, this study nonetheless provides a cornerstone for establishing a potential role of the mTOR pathway in childhood malnutrition. The study findings implicate an evident downregulation in the overall activity of the mTOR pathway to be linked with the pathogenesis of chronic forms of childhood malnutrition. Further studies exploring other integral mTOR pathway components, as well as components of other intracellular pathways known to be interconnected with the mTOR pathway; along with systematic investigations into the well-established upstream modulators of the mTOR pathway, need to be assessed to decipher conclusive roles of the mTOR pathway with childhood malnutrition. Although previous animal model studies have demonstrated an overall down-modulation of the mTOR pathway with reduced cell proliferation and growth, our findings confirm such propositions in humans.

## Figures and Tables

**Figure 1 nutrients-14-01612-f001:**
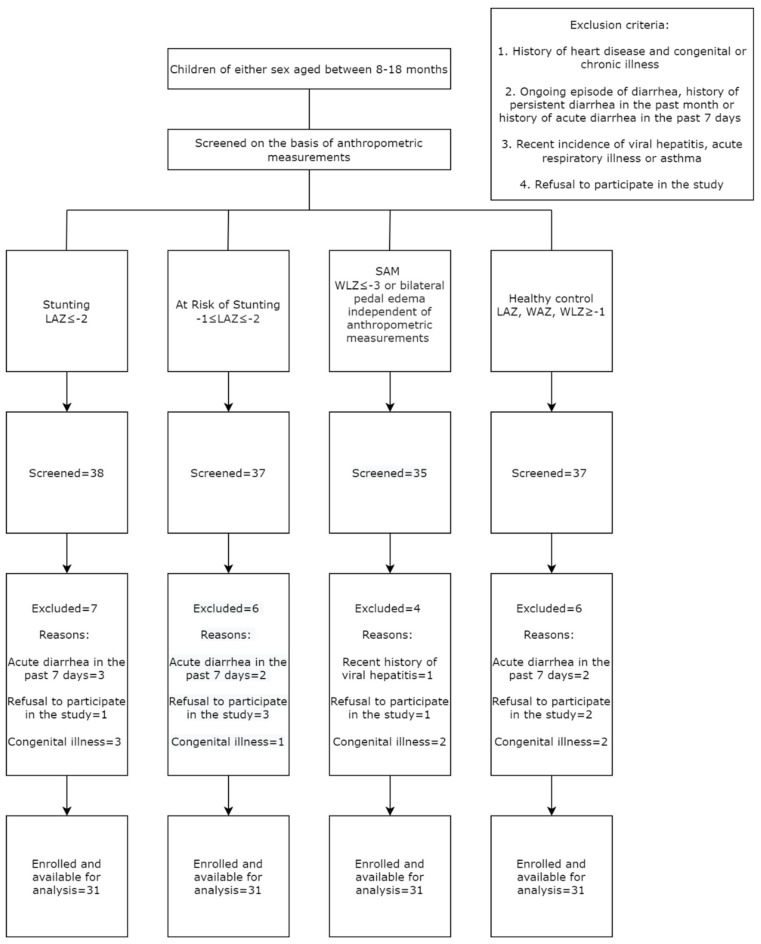
Flow chart showing the inclusion of the participants in the study.

**Figure 2 nutrients-14-01612-f002:**
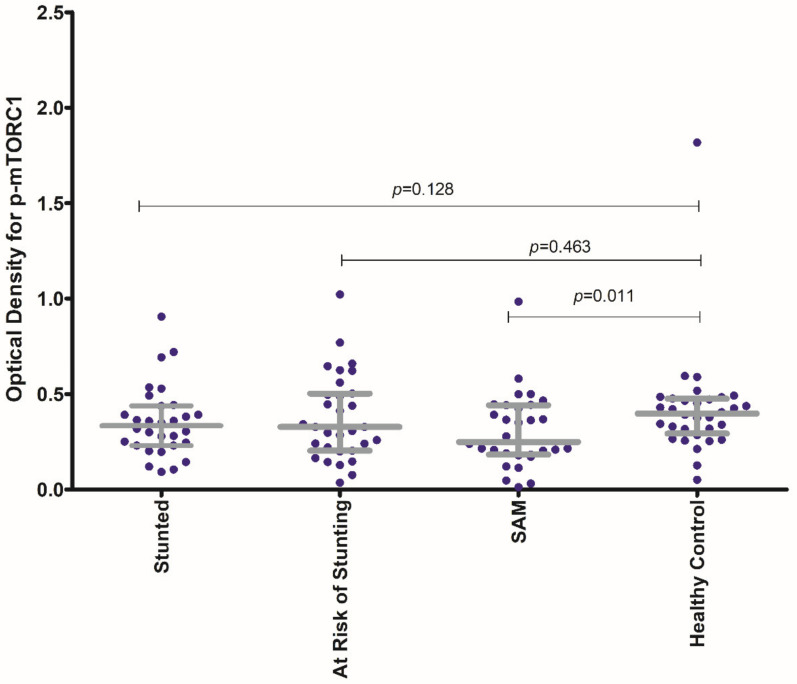
Comparison of the activity of phospho-mTORC1 (as expressed by the OD for phospho-mTORC1 at 450 nm) for each of the malnourished study arms (stunted, at risk of stunting and SAM) with that of the healthy control group. The *p*-value shown is the result of the Mann–Whitney U test comparing the median OD value for p-MTORC1 of each of the malnourished groups (stunted, at risk of stunting, SAM) with that of the healthy control group. The colored dots represent the individual OD values for p-mTORC1 at 450 nm.

**Figure 3 nutrients-14-01612-f003:**
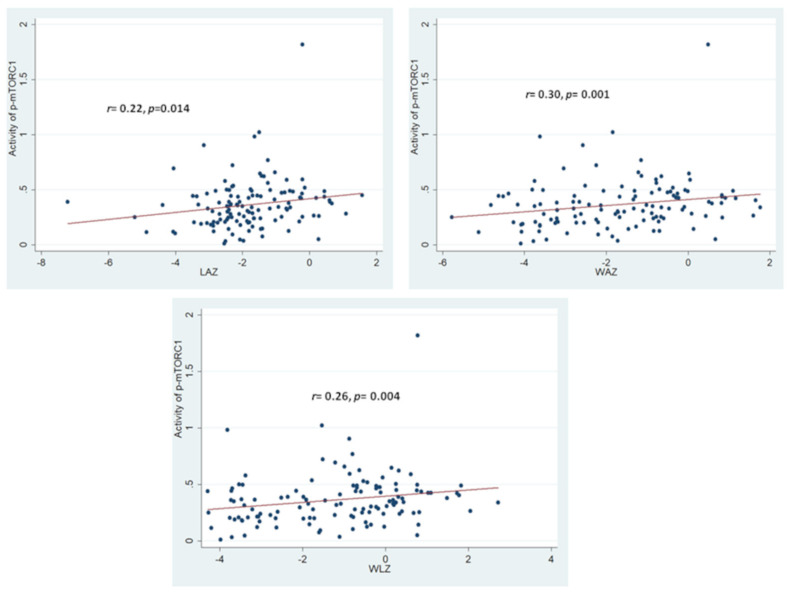
Statistical correlations between activity of p-mTORC1 with the anthropometric indices of LAZ, WAZ and WLZ. Correlation coefficient and the *p*-value shown is the result of Spearman-Rank Correlation between the activity of p-mTORC1 with each of the anthropometric indices. The colored circles indicate the individual activity of p-mTORC1 for the respective anthropometric indices of the study participants.

**Figure 4 nutrients-14-01612-f004:**
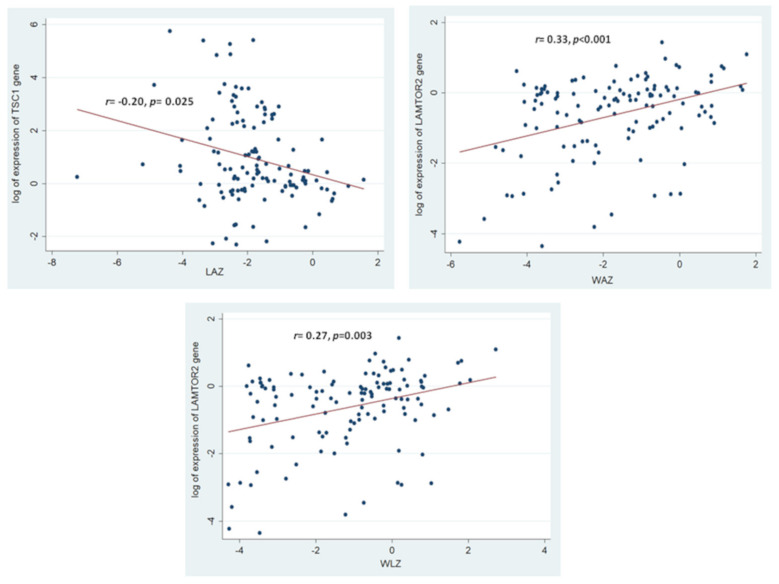
Statistically significant correlations between the expressions of the selected mTOR pathway genes and the anthropometric indices of the study participants. Correlation coefficient and the *p*-value shown are results of Spearman Rank Correlation between the log of expression of the mTOR pathway genes with the anthropometric indices. The colored circles indicate the individual gene expressions for the respective anthropometric indices of the study participants.

**Table 1 nutrients-14-01612-t001:** Socio-demographic, anthropometric and nutritional data collected from the study participants, distributed across the four study arms.

Characteristics	Stunted (*n* = 31)	At Risk of Stunting (*n* = 31)	SAM (*n* = 31)	Healthy Control (*n* = 31)
Age in months	12 (10, 14)	12 (8.5, 15.5)	11 (9, 13)	12 (9.5, 14.5)
Male sex, *n* (%)	16 (51.6%)	20 (64.5%)	19 (61.3%)	16 (51.6%)
Weight in kg	7.2 (6.8, 7.6)	8.15 (7.7, 8.6)	6.51 (6.2, 6.8)	9.82 (8.9, 10.7)
Length in cm	69.5 (67.2, 71.8)	70.3 (66.6, 74)	71.3 (69.4, 73.3)	75.4 (72.5, 78.2)
Mid Upper Arm Circumference in cm	135 (128, 142)	141.5 (136, 147)	120 (118, 123)	150 (144, 156)
Head circumference in cm	43.5 (43, 44)	44.5 (43.3, 45.7)	44 (42.5, 45.5)	44.8 (44, 45.6)
Birth weight in kg	2.5 (2.25, 2.75)	3 (2.55, 3.45)	2.8 (2.55, 3.05)	3.1 (2.70, 3.50)
Weight for age Z score	−2.42 (−2.96, −1.89)	−1.17 (−1.44, −0.9)	−3.75 (−4, −3.54)	0.5 (−0.2, 1.2)
Length for age Z score	−2.35 (−2.6, −2.1)	−1.62 (−1.8, −1.44)	−2.46 (−3.26, −1.67)	−0.22 (−0.76, 0.30)
Weight for length Z score	−1.57 (−2.21, −0.93)	−0.45 (−1.03, −0.13)	−3.66 (−3.81, −3.52)	0.36 (−0.44, 1.16)
Gestational age in weeks	38 (36.8, 39.3)	39 (38.5, 39.5)	38 (37, 39)	39 (38, 40)
Maternal age in years	23 (19.3, 26.8)	22.5 (17, 28)	25 (20, 30)	24.5 (22.4, 26.6)
Duration of EBF in months	3 (0, 6)	1 (0, 3.75)	3 (0, 6)	3 (0, 6)
Currently breastfeeding, *n* (%)	28 (90.3%)	29 (93.6%)	22 (71%)	29 (93.6%)
Currently formula feeding, *n* (%)	15 (48.4%)	16 (51.6%)	29 (93.6%)	19 (61.3%)
Currently fed with rice powder or suji, *n* (%)	14 (45.2%)	16 (51.6%)	28 (90.3%)	14 (45.2%)
Currently fed with cow milk, *n* (%)	9 (29%)	8 (25.8%)	17 (54.8%)	6 (19.4%)
Peripheral blood mononuclear cell count, million cells/ml	4 (2.6, 5.3)	4.35 (1.65, 5.7)	4.75 (3.85, 5.65)	4.33 (2.83, 5.83)

**Table 2 nutrients-14-01612-t002:** Comparison of the expression of the selected mTOR pathway genes (*MTOR*, *TSC1*, *LAMTOR2*, *RPS6K1* and *RICTOR*) between each of the three malnourished cohorts with the healthy control group.

Expression of mTOR Pathway Gene, 2^−∆∆Ct^	Stunted vs. Healthy Control			At Risk of Stunting vs. Healthy Control			SAM vs. Healthy Control		
	Stunted	Healthy Control	*p*-Value	At Risk of Stunting	Healthy Control	*p*-Value	SAM	Healthy Control	*p*-Value
*MTOR*	0.701 (0.233–1.172)	0.882 (0.369–1.389)	0.032	0.549 (0.151–0.963)	0.882 (0.369–1.389)	0.001	0.748 (0.331–1.268)	0.882 (0.369–1.389)	0.088
*TSC1*	1.822 (1.031–2.638)	0.972 (0.733–1.202)	0.016	3.331 (3.331–15.902)	0.972 (0.733–1.202)	0.001	2.062 (2.056–18)	0.972 (0.733–1.202)	0.002
*LAMTOR2*	0.688 (0.298–1.078)	0.991 (0.459–1.522)	0.003	0.823 (0.411–1.221)	0.991 (0.459–1.522)	0.037	0.627 (0.211–1.062)	0.991 (0.459–1.522)	0.007
*RPS6K1*	1.162 (1.162–3.623)	0.904 (0.411–1.381)	0.128	0.968 (0.242–1.713)	0.904 (0.411–1.381)	0.977	1.289 (0.532–2.044)	0.904 (0.411–1.381)	0.438
*RICTOR*	0.731 (0.322–1.131)	0.869 (0.377–1.364)	0.109	0.473 (0.112–0.828)	0.869 (0.377–1.364)	0.013	0.854 (0.472–1.619)	0.869 (0.377–1.364)	0.603

Footnotes: *p*-values are the result of Mann–Whitney U test conducted to compare the gene expression between each malnourished group with that of the healthy control group.

**Table 3 nutrients-14-01612-t003:** Association of the analyzed mTOR pathway components with the different forms of childhood malnutrition; results from multivariate logistic regression analysis.

mTOR Pathway Components Analyzed	Stunted vs. Healthy Control		At Risk of Stunting vs. Healthy Control		SAM vs. Healthy Control	
	Odds Ratio (95% CI)	*p*-Value	Odds Ratio (95% CI)	*p*-Value	Odds Ratio (95% CI)	*p*-Value
phospho-mTORC1	0.102 (0.02–0.803)	0.042	0.063 (0.02–2.162)	0.801	0.202 (0.012–0.822)	0.043
*MTOR* gene	0.377 (0.102–1.453)	0.161	0.153 (0.042–0.639)	0.011	0.203 (0.041–1.073)	0.060
*TSC1* gene	1.622 (1.041–2.513)	0.031	2.509 (1.261–5.011)	0.009	3.151 (0.788–12.511)	0.012
*LAMTOR2* gene	0.111 (0.023–0.494)	0.042	0.188 (0.061–0.612)	0.006	0.204 (0.042–0.974)	0.046
*RPS6K1* gene	1.012 (0.968–1.048)	0.686	0.992 (0.943–1.054)	0.831	0.981 (0.842–1.134)	0.763
*RICTOR* gene	1.063 (0.881–1.282)	0.528	0.788 (0.517–1.223)	0.289	0.514 (0.191–1.412)	0.204

Potential confounders and covariates adjusted in the final regression model included: age, sex, RNA concentration, PBMC count, duration of exclusive breast feeding, current feeding practices (breastfeeding, formula feeding, feeding with rice powder/suji and feeding with cow/goat’s milk), paternal age (for stunted group), maternal age (for stunted group) and maternal occupation (for at risk of stunting group).

## Data Availability

Data related to this manuscript are available upon request and for researchers who meet the criteria for access to confidential data may contact with Armana Ahmed (armana@icddrb.org) of the Research Administration of icddr,b (http://www.icddrb.org/).

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
