# Peer review of "Exploratory Analysis of Selected Components of the mTOR Pathway Reveals Potentially Crucial Associations with Childhood Malnutrition"

_nutrients, 2022, doi:10.3390/nu14081612_

Round 1

Reviewer 1 Report

This analysis seeks to characterize the expression of several elements of a key pathway for growth in animals, the mTOR pathway, among four cohorts of children in Bangladesh: stunted children, those with LAZ between -1 and -2, those with severe acute malnutrition, and a more anthropometrically normal “healthy” cohort. The cohorts totaled ~120 children, who were recruited form either the community or hospital settings in Dhaka. Phosphorylated mTOR and the gene expression of 5 mTOR pathway components (from blood) were analyzed and compared between the groups, with the finding that several mTOR pathway components were expressed at different levels in the abnormal groups vs. controls. These findings are consistent with different mTOR activity in these groups with different anthropometry. Improving understanding of the biological underpinnings of poor growth is very important for child global health, and mTOR analyses holds promise toward this end.

There are several major issues with the manuscript, however. First, the authors do not provide information about the number of children assessed, excluded, etc. in a CONSORT flow diagram. Second, the main findings as displayed in the figures do not effectively aid in understanding. For instance, in figure 2, it is very challenging for the reader to get a sense of the differences between groups on 3 of the 5 panels due to the y axis ranges chosen to include outliers. Relatedly, in table 2 and in figure 1, the “healthy children” data is repeated multiple times for each pairwise comparison; in figure 1, a single panel showing all groups would be better, while in table 2, including the healthy children data a single time, in one column, would make the data easier to digest. Third, the authors should discuss how the mTOR findings relate to specific anthropometric deficits in potentially different manners. LAZ, WAZ, and WLZ are always mentioned in concert for correlations; how do the mTOR elements differ in their expression in relation to LAZ as compared to WLZ? Certainly, stunting and wasting have important differences, and it may be that mTOR elements associate differently in these groups. In addition to analyzing the data on these lines, it would be very helpful to expand the discussion to include this, especially if there do appear to be meaningful differences in mTOR expression between stunted and wasted kids, for instance. Too often, the analysis and discussion lump “malnutrition” together. Relatedly, some data, whether table/figure, on the correlations should be included in the main manuscript, as this is a key point for the manuscript’s message. Perhaps a plot with a series of panels displaying the most important correlations, e.g., TSCI1 expression on x axis and LAZ on y axis. Additionally, none of the logistic regression outputs are listed or discussed in the manuscript at any point, but rather are included in a supplementary table that is not mentioned in the body of the manuscript.

In addition to these issues, there are grammatical errors in the manuscript, as well as phrases repeated in sequence, and some hanging dependent clauses on their own. For example, on page 3 paragraph 2, “Assessment of the presence of bilateral pedal edema… was exclusively by experienced physicians working at the Dhaka hospital of icddr,b. the parents or legal guardians of the participants.” What is the end of this sentence?

Was there Ethics committee approval?

How many SAM children had edema?

When in the process of SAM treatment were these children assessed? Are the measurements in Table 1 at time of blood draw, or admission to hospital?

Table 2 – at several points, the median = Q1, which seems highly unlikely with these continuous data. Please use a consistent number of digits after the decimal point.

There is no mention of the logistic regression analysis outputs in the results or the discussion.

All legends for tables and figures should display the statistical test used to generate p values, as well as describe what is added to the data e.g. median lines, IQR lines, etc.

From animal models we would surmize the mTOR pathway components are very likely to show lower expression in any type of malnutrition.  This study confirms this in humans. This is the main conclusion in my mind.

The 'fix' for mTOR is already known, a diet of ample amino acids and micronutrients.

Author Response

Thank you for your comments on our manuscript. Please kindly find the point-by-point responses to your comments and suggestions.

This analysis seeks to characterize the expression of several elements of a key pathway for growth in animals, the mTOR pathway, among four cohorts of children in Bangladesh: stunted children, those with LAZ between -1 and -2, those with severe acute malnutrition, and a more anthropometrically normal “healthy” cohort. The cohorts totaled ~120 children, who were recruited form either the community or hospital settings in Dhaka. Phosphorylated mTOR and the gene expression of 5 mTOR pathway components (from blood) were analyzed and compared between the groups, with the finding that several mTOR pathway components were expressed at different levels in the abnormal groups vs. controls. These findings are consistent with different mTOR activity in these groups with different anthropometry. Improving understanding of the biological underpinnings of poor growth is very important for child global health, and mTOR analyses holds promise toward this end. There are several major issues with the manuscript, however.

Comment: First, the authors do not provide information about the number of children assessed, excluded, etc. in a CONSORT flow diagram.

Response: Thank you for your comment. We have added a participant flow chart in the revised version of the manuscript. Please kindly refer to Figure 1 of the revised manuscript.

Comment: Second, the main findings as displayed in the figures do not effectively aid in understanding. For instance, in figure 2, it is very challenging for the reader to get a sense of the differences between groups on 3 of the 5 panels due to the y axis ranges chosen to include outliers. Relatedly, in table 2 and in figure 1, the “healthy children” data is repeated multiple times for each pairwise comparison; in figure 1, a single panel showing all groups would be better, while in table 2, including the healthy children data a single time, in one column, would make the data easier to digest.

Response: Thank you for your comment. We have redone the figure showing pairwise comparison of p-mTORC1 between each of the three malnourished groups with that of the healthy control group. Please kindly refer to Figure 2 of the revised version of the manuscript. Moreover, we have omitted Figure 2 of the previous version of the manuscript, since Table2 illustrates the same data. However, we were unable to change the style of data representation of Table 2 in the revised version, since we believe that pairwise comparison of each of the gene expressions is more clearly represented by such data representation.

Comment: Third, the authors should discuss how the mTOR findings relate to specific anthropometric deficits in potentially different manners. LAZ, WAZ, and WLZ are always mentioned in concert for correlations; how do the mTOR elements differ in their expression in relation to LAZ as compared to WLZ? Certainly, stunting and wasting have important differences, and it may be that mTOR elements associate differently in these groups. In addition to analyzing the data on these lines, it would be very helpful to expand the discussion to include this, especially if there do appear to be meaningful differences in mTOR expression between stunted and wasted kids, for instance. Too often, the analysis and discussion lump “malnutrition” together. Relatedly, some data, whether table/figure, on the correlations should be included in the main manuscript, as this is a key point for the manuscript’s message. Perhaps a plot with a series of panels displaying the most important correlations, e.g., TSCI1 expression on x axis and LAZ on y axis. Additionally, none of the logistic regression outputs are listed or discussed in the manuscript at any point, but rather are included in a supplementary table that is not mentioned in the body of the manuscript.

Response: Thank you for your comment. We have added a paragraph in the Discussion section of the revised manuscript, where we have attempted to explain for the relationship between the anthropometric deficits with the difference in activity of the mTOR pathway components between each of the malnourished groups with that of the healthy control group. We have displayed the significant statistical correlations between the activity of the mTOR pathway components with the anthropometric indices of LAZ, WAZ and WLZ (Figure 3 and Figure 4 of the revised manuscript. We have also added a table showing the results of the regression analysis in the Results section of the revised version of the manuscript (Table 3).

Comment: In addition to these issues, there are grammatical errors in the manuscript, as well as phrases repeated in sequence, and some hanging dependent clauses on their own. For example, on page 3 paragraph 2, “Assessment of the presence of bilateral pedal edema… was exclusively by experienced physicians working at the Dhaka hospital of icddr,b. the parents or legal guardians of the participants.” What is the end of this sentence?

Response: Thank you for your comment. We have addressed the grammatical errors and discrepancies mentioned by the reviewer.

Comment: Was there Ethics committee approval?

Response: Thank you for your comment. Yes, due approval was obtained from the ethical review committee of icddr,b, prior to the start of the study.

Comment: How many SAM children had edema?

Response: Thank you for your comment. Only one child in the SAM group had bilateral pedal edma

Comment: When in the process of SAM treatment were these children assessed? Are the measurements in Table 1 at time of blood draw, or admission to hospital?

Response: Thank you for your comment. The SAM children were enrolled after acute phase management and prior to start of nutritional rehabilitation. All data collected were after acute phase management and also blood was drawn after acute phase management and prior to start of nutritional rehabilitation.

Comment: Table 2 – at several points, the median = Q1, which seems highly unlikely with these continuous data. Please use a consistent number of digits after the decimal point.

Response: Thank you for your comment. The data for gene expression of the mTOR pathway genes was skewed and so we used median and the first and third quartiles to represent the data. In the revised manuscript, we have used 3 digits after decimal point for representation of all data in Table 2.

Comment: There is no mention of the logistic regression analysis outputs in the results or the discussion.

Response: Thank you for your comment. We have added a table showing the results of the regression analysis in the Results section of the revised version of the manuscript (Table 3).

Comment: All legends for tables and figures should display the statistical test used to generate p values, as well as describe what is added to the data e.g. median lines, IQR lines, etc.

Response: Thank you for your comment. The legends of the revised figures to display the required information.

Comment: From animal models we would surmize the mTOR pathway components are very likely to show lower expression in any type of malnutrition.  This study confirms this in humans. This is the main conclusion in my mind.

Response: Thank you for your comment. We have modified our conclusion accordingly.

Comment: The 'fix' for mTOR is already known, a diet of ample amino acids and micronutrients.

Response: Thank you for your comment. We have added this information in the discussion of the revised manuscript.

Reviewer 2 Report

The authors need to check the manuscript carefully as some of the English is a little odd or there are missing words - e.g. abstract 4 lines from the end . Our findings provide key insights possible down-modulation .. should be INTO possible. 

Introduction - change 'till date' to 'until now'

Materials and Methods 2.2 there is a repeat of the parents or legal guardians of the participants.  Check the sample size formula currently incorrect

Author Response

Dear sir, 

Thank you for your kind review of our work. Please find below the point-by-point responses to your suggestions and comments. 

Comment: The authors need to check the manuscript carefully as some of the English is a little odd or there are missing words - e.g. abstract 4 lines from the end . Our findings provide key insights possible down-modulation .. should be INTO possible. 

Response: Thank you for the comment. We have checked the manuscript thoroughly and have corrected the above-mentioned grammatical errors.

Comment: Introduction - change 'till date' to 'until now'

Response: Thank you for the comment. We have made the above-mentioned change as suggested by the reviewer.

Comment: Materials and Methods 2.2 there is a repeat of the parents or legal guardians of the participants.  Check the sample size formula currently incorrect

Response: Thank you for the comment. We have corrected the discrepancy and have removed the repetition. As suggested by the reviewer, we have also checked the sample size formula and have corrected it.

Round 2

Reviewer 1 Report

I thank the authors for their revisions.